# The Influence of the Production Process on the Anthocyanin Content and Composition in Dried Potato Cubes, Chips, and French Fries Made from Red-Fleshed Potatoes

Elżbieta Rytel [1],*, Agnieszka Tajner-Czopek [1], Agnieszka Kita [1], Agnieszka Tkaczyńska [1], Alicja Z. Kucharska [2] and Anna Sokół-Łętowska [2]

1   Department of Food Storage and Technology, Wrocław University of Environmental and Life Sciences, Chełmońskiego 37 Str., 51-630 Wrocław, Poland; Agnieszka.tajner-czopek@upwr.edu.pl (A.T.-C.); Agnieszka.kita@upwr.edu.pl (A.K.); agnieszka.tkaczynska@upwr.edu.pl (A.T.)
2   Department of Fruit, Vegetable and Plant Nutraceutical Technology, Wrocław University of Environmental and Life Sciences, Chełmońskiego 37/41 Str., 51-630 Wrocław, Poland; Alicja.kucharska@upwr.edu.pl (A.Z.K.); anna.sokol-letowska@upwr.edu.pl (A.S.-Ł.)
*   Correspondence: elzbieta.rytel@upwr.edu.pl; Tel.: +48-713207769

**Abstract:** The stability of acylated anthocyanins is still a new and unexplored subject of study. The changes in the contents of individual anthocyanins in colored-flesh potato tubers during processing have rarely been addressed in the literature. The aim of the present study was to determine how anthocyanin degradation and profiles are influenced in potatoes of the red-fleshed Herbie 26 variety by different methods of processing. Potato samples were divided into four categories to be analyzed, namely, raw material, potato cubes, French fries, and chips. The dried cubes, French fries, chips, semi-finished products, and finished products, obtained through laboratory processing, were examined for anthocyanin content and composition. The production process of cubes, chips, and French fries led to losses of the examined anthocyanins; however, these losses differed depending on the technological stage. The greatest losses of these compounds were determined after the final production processes, i.e., pre-frying, frying, and drying. Chip production led to the lowest losses of anthocyanins. Omitting the blanching stage in chip production allowed the retention of more anthocyanins. Pelargonidin-3-feruloylrutinoside-5-glucoside, having the highest percentage in the raw material (approximately 50%), followed by pelargonidin-3-caffeoylrutinoside-5-glucoside, proved to be the most thermally stable.

**Keywords:** red-fleshed potatoes; potato products; anthocyanin content; stability of acylated anthocyanins

## 1. Introduction

Potatoes are one of the most popular raw materials worldwide, readily available in most countries and widely consumed compared to other root vegetables, such as carrots, parsley, or celery. However, in highly industrialized countries, potato consumption has decreased in favor of processed potato products (e.g., French fries and chips) and semi-finished potato products (e.g., dried potato cubes). Over the last two decades (2000–2019), the consumption of unprocessed potatoes in Poland has decreased by approximately 31%, while the consumption of potato products has increased by the same percentage [1]. As consumers are becoming more discerning, the food industry has been striving to create and market new, more attractive product ranges to boost sales. Recently, products made from red- and purple-fleshed potatoes have been introduced in many countries. Though they are still less popular than those made from traditional, light-fleshed varieties, a growing number of conscious consumers are opting for healthy food products, attractive in terms of appearance, shape, color, and flavor.

The advantage of red- and purple-fleshed potato varieties lies not only in their attractive color, but also in their higher nutritional value compared to traditional light-fleshed varieties [2,3]. Colored-flesh potatoes are characterized by an almost two-to-three times higher content of polyphenolic compounds and higher antioxidant activity than light-fleshed (yellow) ones [4,5]. This is, to a large extent, due to anthocyanins—natural pigments found in the colored varieties. Potato anthocyanins include acetylated cyanidin derivatives, with colors ranging from orange to red, blue, and violet, depending on the pH of the environment [4,6,7]. Potatoes contain water-soluble anthocyanins, which are stable in acidic environments, including pigments isolated from fruit, as well as in neutral or slightly alkaline environments [8,9]. Plant pigments are very unstable, and their activity can change depending on pH, temperature, presence of metal ions, or oxygen [10,11]. Processing reduces the content of polyphenolic compounds, including anthocyanins, in plant raw materials, while also changing their activity and bioavailability. This process is very complex and highly variable depending on the type of raw material and production technology used, so it is up to food producers to retain as many of the health-promoting compounds as possible. Therefore, it is important to monitor the content of health-promoting compounds and changes that occur under the influence of food processing factors.

The stability of acylated anthocyanins is still a new and unexplored subject of study. The changes in the contents of individual anthocyanins in colored-flesh potato tubers during processing have rarely been addressed in the literature. Therefore, the aim of the present study was to determine how anthocyanin degradation and profiles are influenced in potatoes of the red-fleshed Herbie 26 variety by different methods of processing.

## 2. Materials and Methods

### 2.1. Plant Material

Red-fleshed potatoes of the Herbie 26 variety served as the material for this study. Potatoes were supplied by the Research Station of the Institute of Agroecology and Plant Production of the Wrocław University of Environmental and Life Sciences, and were harvested in 2016, 2017, and 2018. The potato samples were then divided into four categories to be analyzed, namely, raw material, potato cubes, French fries, and chips.

The variables measured in the raw material included the contents of dry matter, starch, total and reducing sugars, as well as anthocyanin composition and total anthocyanin content. The dried cubes, French fries, chips, semi-finished products (slices, blanched raw material, fries after one-stage frying), and finished products, obtained through laboratory processing, were examined for dry matter content and anthocyanin content and composition.

### 2.2. Production of Dried Potato Cubes, French Fries, and Chips

The dried potato cubes were made from peeled tubers (manually peeled with a vegetable peeler), which were cut into $1 \times 1 \times 1$ cm cubes using a slicer, blanched in water at 75 °C for 5 min, pre-dried at 120 °C for 1 h, and further dried at 55–60 °C in a ventilated desiccator until reaching a 10–12% moisture content. The French fries were prepared by peeling the tubers (as described above), then cutting them into $1 \times 1 \times 10$ cm strips with a slicer (Robot Coupe CL 50, Technica Group Sp. z o.o., Skoczów, Poland), blanching in water at 75 °C for 5 min, pre-drying at 30 °C in a ventilated desiccator, and two-stage frying in rapeseed oil at 175 °C. The pre-dried raw material was pre-fried at 175 °C for 1 min, then fried at 175 °C for 5 min in a laboratory fryer (RM Gastro, "ULM-NEU-ULM", Zielona Góra, Poland). The chips were obtained by peeling the potatoes as described above and cutting them into 1 mm slices with a slicer. The potato slices were rinsed in water and dried on a paper towel, then fried for approximately 2 min at two different temperatures: 150 and 170 °C.

### 2.3. Analysis

#### 2.3.1. Sample Preparation

The potato samples were washed and cut into small pieces and freeze-dried (using an Edwards Modulyo 4KII freeze dryer, West Sussex, UK). The dry material was ground in an electric grinder to a fine powder and used for the quantification and identification of anthocyanins.

#### 2.3.2. Extraction of Anthocyanins

The samples were prepared according to the method described by Nemś et al. [12]. Dry samples (1.001 mg), e.g., of raw potatoes or potato products, were extracted with 70% aqueous acetone (0.1% acetic acid). The mixture was prepared in a graduated tube and then homogenized using a vortex. After thorough mixing, the sample was left for 2 h at room temperature. The lipophilic compounds present in the samples (acetone–water) were removed with chloroform. The acetone–water fraction was collected and evaporated using a Büchi rotary evaporator (Merck, Darmstadt, Germany). The obtained extract was collected with 50% methanol to a known volume and was then stored at room temperature prior to analysis. Next, the samples were filtered with 0.45 and 0.22 μm filters and analyzed by HPLC–PDA.

#### 2.3.3. Quantification of Anthocyanins by HPLC–PDA

The content of anthocyanins was determined according to Kucharska et al. [13] using a Dionex HPLC (Walthman, MA, USA) system equipped with an Ultimate 3000 model of a diode array detector, an LPG-3400A quaternary pump, an EWPS-3000SI auto sampler, and a TCC-3000SD thermostatted column compartment, controlled by Chromeleon v.6.8 software. The Cadenza Imtakt column C5–C18 (75 × 4.6 mm, 5 μm) was used. The following solvents constituted the mobile phase: 4.5% formic acid (solvent A) and 100% acetonitrile (solvent B). The following elution conditions were applied: 0–1 min 5% B in A; 1–20 min 25% B in A; 20–27 min 100% B in A; 27–30 min 5% B in A. The flow rate was 1 mL/min, and the injection volume was 40 μL. The column was operated at 30 °C. Anthocyanins were monitored at 520 nm and their content was expressed in cyanidin 3-*O*-glucoside equivalents (CygE)/100 g d.m (dry matter). The calibration curve was obtained by the external standard method on six concentration levels (10–75 μg/mL) of standard compound (cyanidin 3-*O*-glucoside—Extrasynthese), with three injections per level. Chromatogram peak areas were plotted against the known concentrations of the standard solutions to establish the calibration equation (y = 0.8538x, where y is the concentration and x is the area). A linear regression equation was calculated by the least squares method. As the regression coefficient ($R^2$) was 0.9994, the relationship was considered linear, and thus acceptable for quantifying the compound.

#### 2.3.4. Identification of Anthocyanins by Ultra-Performance Liquid Chromatography (UPLC)–Q-TOF–MS/MS

The method for anthocyanin identification was previously described by Mizgier et al. [14]. Anthocyanin pigments were analyzed using an Acquity ultra-performance liquid chromatography (UPLC) system combined with a PDA detector and connected to a quadrupole-time of flight (Q-TOF) MS instrument (UPLC/Synapt Q-TOF MS, Waters Corp., Milford, MA, USA) with an electrospray ionization source (ESI). They were separated on an Acquity TM BEH C18 column (100 × 2.1 mm i.d., 1.7 μm; Waters, Merck, Darmstadt, Germany). The mobile phase was a mixture of 4.5% formic acid (solvent A) and 100% acetonitrile (solvent B). The gradient program was as follows: Initial conditions, 99% (A); 12 min, 75% (A); 12.5 min, 100% (B); 13.5 min, 99% (A). The column was operated at 30 °C, the flow rate was 0.45 mL/min, and the injection volume was 5 μL. The detection wavelength was set to 520 nm.

The main operating parameters for the Q-TOF MS were set as follows: Cone voltage, 40 V; capillary voltage, 2.0 kV; cone gas flow, 11 L/h; dissolution temperature, 250 °C;

source temperature, 100 °C; collision energy, 28–30 eV; collision gas, argon; dissolution gas, nitrogen; flow rate, 600 L/h; ionization mode, positive; data acquisition range, *m/z* 100–2000 Da. The data were collected using Mass-Lynx TM V 4.1. software.

### 2.4. Analytical Methods

The dry matter content of the fresh potato samples and the freeze-dried material was determined by the reduced-weight method after drying at 105 °C until a constant weight had been achieved [15]. The contents of total and reducing sugars and of starch were determined in raw potato tubers with the standard chemical method [16] (Table 1), whereas the content and identification of anthocyanins were analyzed with the methods of Nemś et al. [12], Mizgier et al. [14], and Kucharska et al. [13]. All analyses were carried out in triplicate.

**Table 1.** Chemical compounds of the raw potatoes.

| Chemical Compounds | Potato Variety |
|---|---|
| Dry matter (g/100 g f.m.) | $22.5 \pm 0.12$ |
| Starch (g/100 g f.m.) | $15.2 \pm 0.11$ |
| Total sugar (g/100 g f.m.) | $0.55 \pm 0.09$ |
| Reducing sugar (g/100 g f.m.) | $0.24 \pm 0.08$ |
| Anthocyanins (mg/100 g f.m.) | $6.62 \pm 0.11$ |

The value after "$\pm$" represents the standard deviation (SD); *n* = 6; f.m.—fresh mass.

### 2.5. Statistical Analysis

Statistical calculations were performed using Statistica 13.0 software (StatSoft Polska Sp. Z o.o., Kraków, Poland). Data were subjected to multi-way analysis of variance and the significance of differences between the mean values was determined by Duncan's test ($p \leq 0.05$). All experiments were performed in three technological replications from three years of investigations and data are reported as mean $\pm$ standard deviation (SD) of all data combined.

## 3. Results and Discussion

The examined variety of red-fleshed potatoes (Herbie 26) can be recommended as a raw material for the production of French fries, chips, and dried potato cubes. The usefulness of tubers for potato processing is determined, to a great extent, by the contents of dry matter, starch, and—most importantly—reducing sugars. The examined potatoes had an optimal composition, i.e., 22.5% dry matter, 15.2% starch, and 0.24% reducing sugars. A high content of reducing sugars causes unfavorable changes (browning) in high-temperature-treated food. This is problematic due to the unattractive color of the finished fries, chips, or dehydrated cubes. Herbie 26 potatoes had a total content of cyanidin-3-*O*-glucoside at 6.62 mg/100 g f.m. (Table 2).

**Table 2.** Mass spectrometric properties of the anthocyanins found in the studied colored-flesh potatoes.

| | Pic No. | Compounds | [MS]$^+$ (*m/z*) | MS/MS (*m/z*) |
|---|---|---|---|---|
| **Red-fleshed potato** | 1. | Pelargonidin-3-rutinoside-5-glucoside | 741.242 | 271.070/433.113/579.169 |
| | 2. | Pelargonidin-3-rutoside | 579.162 | 271.070/433.113 |
| | 3. | Pelargonidin-3-caffeoylrutinoside-5-glucoside | 903.248 | 271.070/433.113/741.208 |
| | 4. | Pelargonidin-3-*p*-coumarolrutinoside-5-glucoside | 887.259 | 271.070/433.113/725.190 |
| | 5. | Pelargonidin-3-feruoylrutinoside-5-glucoside | 917.273 | 271.070/433.113/741.208 |

According to Ezekiel et al. [17], in red-fleshed varieties of potatoes, the content of anthocyanins usually exceeds 7 mg/g f.m. In turn, Tierno et al. [18] concluded that the Highland Burgundy Red and Purple Peruvian varieties were the richest in anthocyanins among the colored-flesh potato varieties tested, followed by Rogue de Flanders, Violet Quenn, and Vitelotte. In general, red-fleshed varieties contain fewer anthocyanins than purple-fleshed ones; however, the intensity of the tuber color is also a factor. Uniformly colored potatoes contain more pigments than those with light (yellow or white) spots [19,20]. Pelargonidin-3-feruloylrutinoside-5-glucoside was the major identified anthocyanidin, accounting for 50% of the identified anthocyanidins (Figure 1).

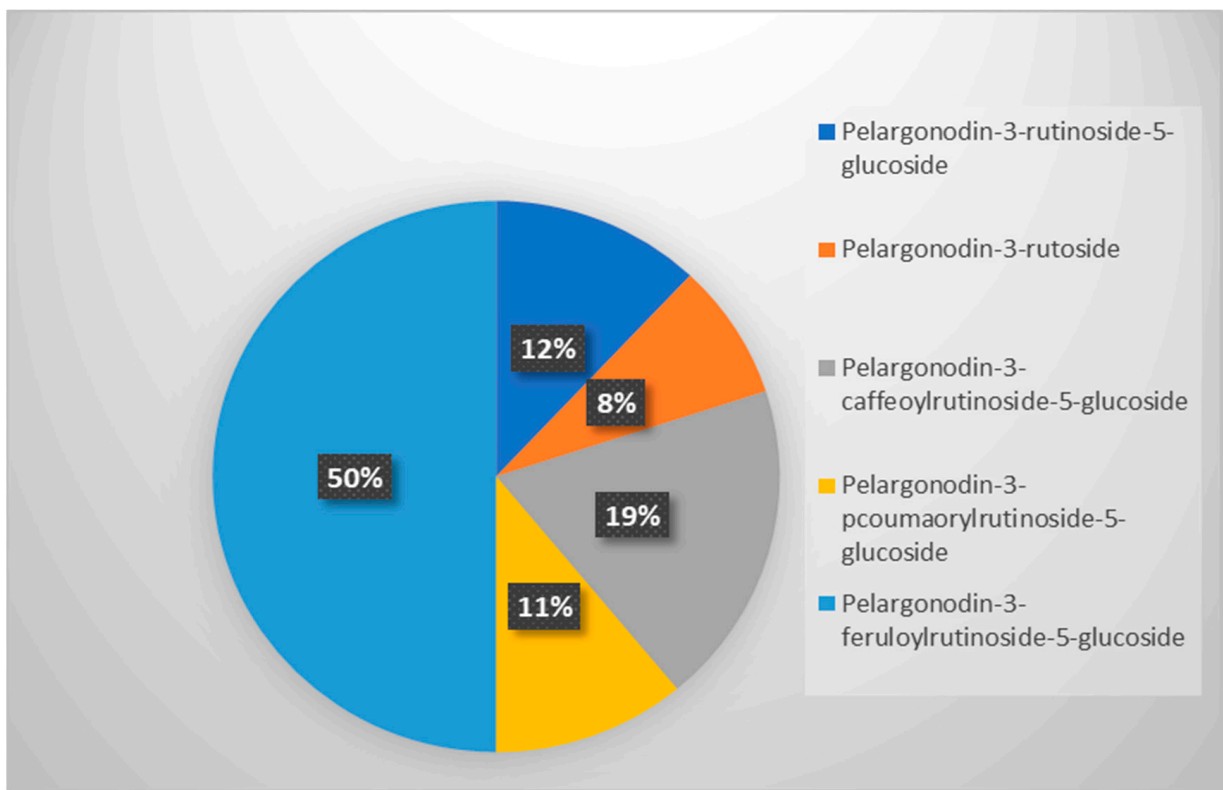

**Figure 1.** Share of the various anthocyanins in the raw potatoes.

According to Lewis et al. [21], red-fleshed potatoes contain mainly pelargonidin and peonidin, whereas purple-fleshed varieties contain mainly petunidin and malvidin.

One of the problems of using anthocyanins as natural dyes in food production is their low stability. Another problem is that most of the raw materials or semi-finished plant products containing natural dyes undergo thermal processes during their processing. Such processes include blanching, cooking, pasteurization, frying, and baking, and according to Jil et al. [22], as a result of the process of pomegranate juice pasteurization, anthocyanin losses of 8–14% occur.

Herein, the production processes used, such as blanching, pre-drying, frying, and drying, had an adverse effect on the content of health-promoting compounds, including pigments, in the raw material and semi-finished products (Figure 2). Processing the potatoes into French fries, chips, and dried potato cubes changed the percentages of the individually identified anthocyanins.

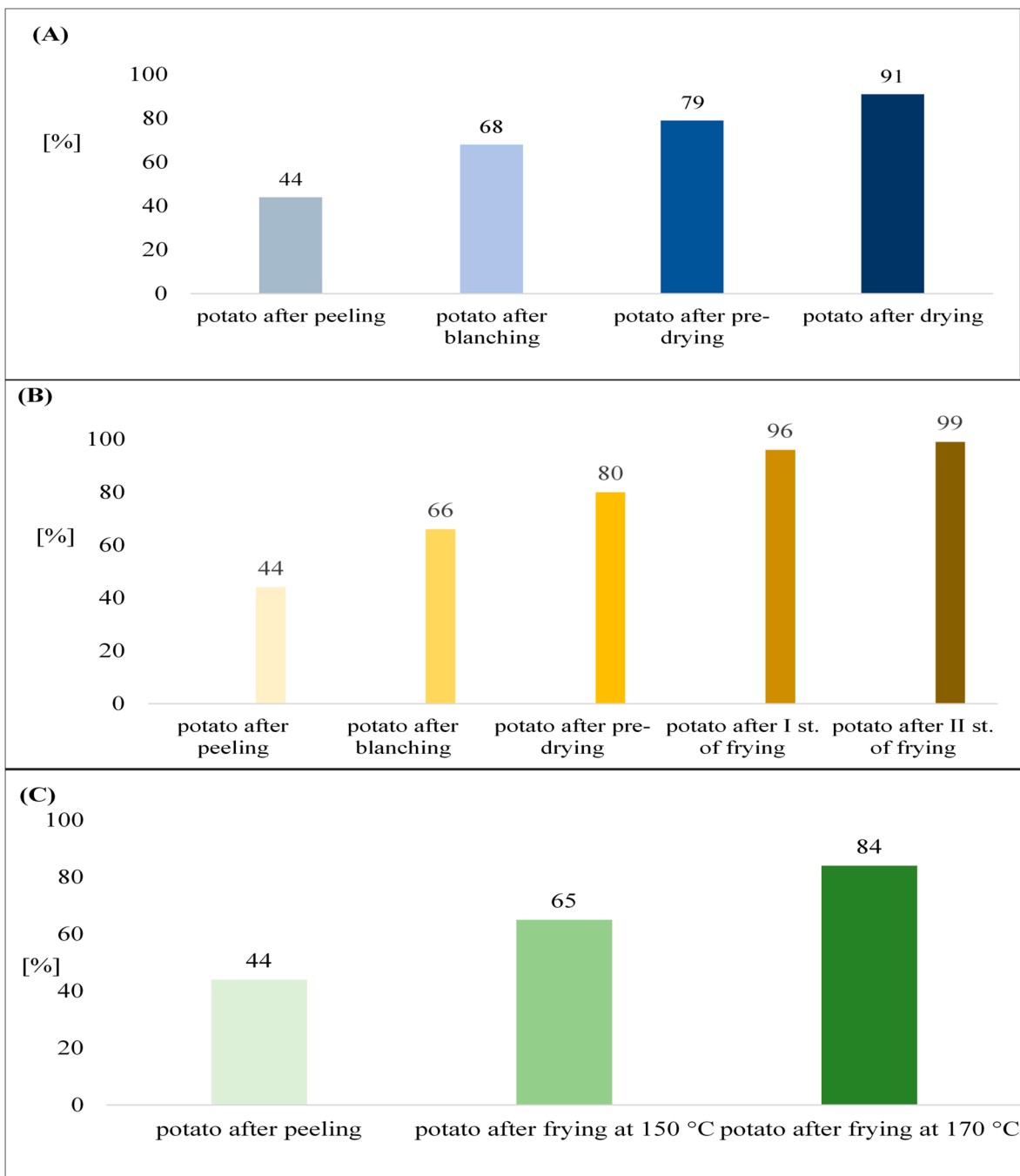

**Figure 2.** Increasing loss of anthocyanin content (%) in potatoes after technological stages of the processing of dried potato cubes (**A**). Increasing loss of anthocyanin content (%) in potatoes after the technological stages of the processing of French fries (**B**). Increasing loss of anthocyanin content (%) in potatoes after the technological stages of the processing of chips (**C**).

According to Lachman et al. [19], the degree of anthocyanin degradation under thermal factors is determined not only by their content in raw materials, but also by the pH process. They concluded that a pH of 3 ensures the best results in terms of anthocyanin retention and that the anthocyanin content in the finished product is largely correlated with the initial content in potatoes and mainly determined by the variety used. By contrast, McDougall et al. [23] stated that the amount of anthocyanins retained is linked to the blanching regime, with anthocyanin losses increasing with blanching time. These authors stated that blanching times of 2–5 min contribute to the preservation of anthocyanins in raw materials, whereas longer temperature treatments increase anthocyanin losses.

However, most authors [8,10,24] have found that these compounds are highly unstable at high temperatures. Moreover, they reported that anthocyanin losses are influenced by enzymatic oxidation processes, which occur in raw materials during cutting, crushing, or extraction. The thermal processes used in the present study are commonly used in the potato industry. Peeling was the first stage in this study, in which the potatoes were peeled manually with a vegetable peeler to a depth of over 1.5 mm, with some potato flesh removed with the peel. The process resulted in anthocyanin losses of 44%, with pelargonidin-3-caffeorylrutinoside-5-glucoside being the most affected (69%), and with pelargonidin-3-rutinoside-5-glucoside and pelargonidin-3-rutoside proving to be the most resistant (31% and 32%, respectively) (Figure 3).

According to Friedman [25], such polyphenolic compounds as anthocyanins and phenolic acids are mainly located in the skin and the tissues directly under it, and thus, peeling may reduce their content in the tubers. Furrer [26] reported that the anthocyanin losses in the food industry due to peeling are approximately 7%, whereas Lachman et al. [27] stated that peeling potatoes to depths even as high as 2 mm does not affect the anthocyanin content. After peeling, the potatoes were diced, cut into strips or sliced (depending on the process), then rinsed with water to remove the starch released from the damaged potato cells or the excess reducing sugars. The cut potatoes were then blanched, though this step was omitted in the production process of chips, as it resulted in an inferior texture of the chips after frying (the finished product absorbed more fat and had a greasy consistency). Blanching is particularly discouraged if the potato tubers already have the optimal content of reducing sugars, while the potatoes used for making dried cubes and fries were blanched in water at 75 °C for 5 min. After this stage, the anthocyanin losses were found to be more significant in finely cut (diced) potatoes, reaching 68% compared to the losses of 66% found in the potato strips (Figure 2). Among the identified anthocyanidins, the greatest losses were recorded for pelargonidin-3-*p*-coumaroylrutinoside-5-glucoside (97%) in dried potato cube and pelargonidin-3-caffeorylrutinoside-5-glucoside (69%) in French fries (Figure 3). The high reduction of anthocyanin content after blanching may have been caused by enzymatic oxidation processes in the raw material during cutting. The activity of o-diphenol oxidase or peroxidase coupled with the effects of polyphenolic compounds leads to a darkened (browning) or colorless product [28,29]. These processes irreversibly impair the organoleptic qualities of food. High-temperature processes (at over 100 °C) led to the most significant anthocyanin losses. For the production of dried potato cubes, prior to proper drying, the cut potatoes were pre-dried at 120 °C for approximately 2 h and then dried at 50 °C for 8 h. This technological stage is very time-consuming, with long pre-drying times necessitated by the difficulty of removing water from the material. Raw material placed in a drying room has an approximately 80% moisture content, which is reduced to 10–12% by drying. Thus, extensive dehydration of raw material is difficult and requires long drying times. The drying process used in the potato industry requires similar parameters: The cut potatoes are pre-dried at approximately 120 °C for around 2 h and then further dried at approximately 80 °C, gradually decreasing the temperature to 50–60 °C for approximately 6–8 h [30]. The long temperature treatment of the cut raw material contributed to large anthocyanin losses and changes in anthocyanin percentages. The highest reductions were shown for pelargonidin-3-rutinoside-5-glucoside, pelargonidin-3-rutoside, and pelargonidin-3-p-coumaroylrutinoside-5-glucoside, whereas the compounds most resistant to the technological processes were pelargonidin-3-feruloylrutinoside-5-glucoside and, to a lesser extent, pelargonidin-3-caffeoylrutinoside-5-glucoside (Figure 3). The frying process is much shorter than drying but is conducted at higher temperatures (over 175 °C), which leads to the degradation of heat-labile compounds. After the second stage of frying, the highest losses were shown for pelargonidin-3-rutoside, while the lowest were shown for pelargonidin-3-rutinoside-5-glucoside, pelargonidin-3-caffeoylrutoside-5-glucoside, and pelargonidin-3-p-coumaroylrutinoside-5-glucoside, though differences in the losses of all of the tested anthocyanins were in the 2–3% range (Figure 2). Different frying temperatures were used to prepare the chips: A lower temperature of 150 °C and a higher temperature

of 170 °C are traditionally used in the industry. The chips fried at the lower temperature showed smaller losses of the examined anthocyanins (65%). By contrast, frying at 170 °C led to anthocyanin losses of 84%, which were still approximately 10% lower than in the production of dried potato cubes or French fries. As with the dried potato cube process, higher stability during chip production was shown for pelargonidin-3-feruloylrutinoside-5-glucoside and pelargonidin-3-caffeoylrutinoside-5-glucoside (Figure 3).

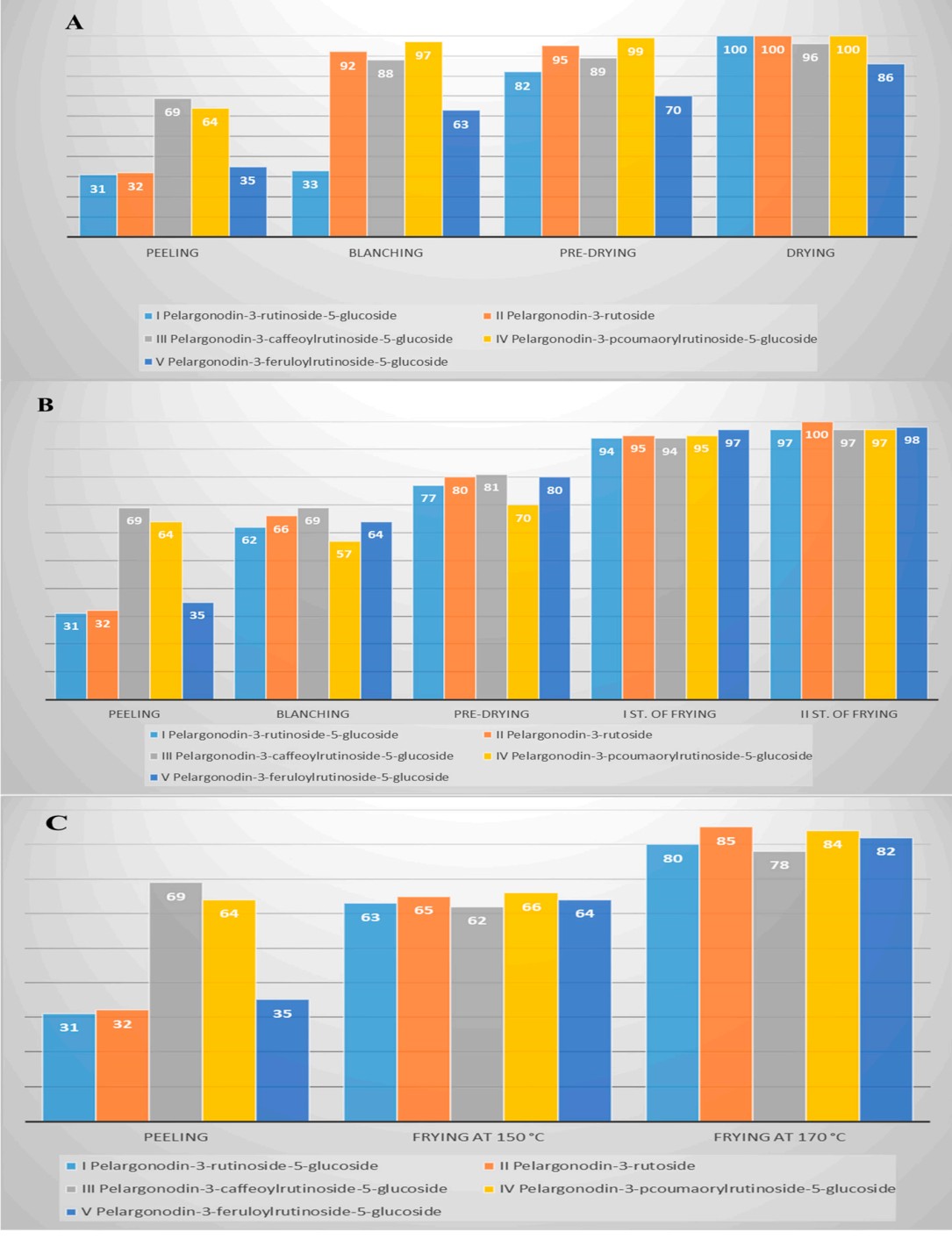

**Figure 3.** Increasing loss of anthocyanin content (%) in potatoes after the technological stages of the processing of dried potato cubes (**A**). Increasing loss of anthocyanin content (%) in potatoes after the technological stages of the processing of French fries (**B**). Increasing loss of anthocyanin content (%) in potatoes after the technological stages of the processing of chips (**C**).

The dried cubes retained 9% of the initial anthocyanin content in the raw material, while the pre-frying and frying processes contributed to higher losses (Figures 2 and 3). The finished French fries contained only 1% of the initial amount of anthocyanins in the potatoes (Figure 3), while chip production led to the lowest anthocyanins losses. The chips fried at the lower temperature (150 °C) retained 35%, whereas those fried at 170 °C retained 16% of the initial anthocyanin content in the raw material (Figure 2). Some studies [31,32] recommend using co-pigments or raw materials containing phenolic acids, which exhibit a protective effect on anthocyanins. In the future, certain modifications should be made to the production processes to reduce losses of natural health-promoting compounds, including pigments, from the raw material, e.g., by lowering the pH of the blanching bath or using lower temperatures for heat treatment (Figure 3). However, not all compounds that form links with anthocyanins favor their stability. Examples include sugars and their degradation products. According to Cevallos-Casals et al. [33], the rate of anthocyanin transformation depends on the degree of sugar degradation into furfural-type compounds that are formed, e.g., during the Maillard reaction.

## 4. Conclusions

Based on the present study, it was found that potatoes of the Herbie 26 variety can be recommended for the production of dried potato cubes, chips, and French fries due to the optimal chemical composition of tubers, particularly concerning their low reducing sugar content (0.24%). Pelargonidin-3-feruloylrutinoside-5-glucoside was the major anthocyanin identified (50%). The production process of dried potato cubes, chips, and French fries led to losses of the examined anthocyanins; however, these losses differed depending on the technological stage. The greatest losses of these compounds were determined after the final production processes, i.e., pre-frying, frying, and drying. The dried potato cubes retained 9% of the initial anthocyanin content, whereas this percentage was only 1% in French fries. Chip production led to the lowest anthocyanin losses. The chips fried at the lower tested temperature retained 35%, whereas the high-temperature-fried ones retained 16% of the initial anthocyanin content. Omitting the blanching stage in chip production allowed the retention of more anthocyanins. Pelargonidin-3-feruloylrutinoside-5-glucoside, having the highest percentage in the raw material (approximately 50%), followed by pelargonidin-3-caffeoylrutinoside-5-glucoside, proved to be the most thermally stable. Losses of the other anthocyanins at the final processing stage were similar and ranged from approximately 2% to 3%.

**Author Contributions:** Conceptualization, E.R.; investigation, E.R. and A.T.; methodology, A.Z.K. and A.S.-Ł.; supervision, A.T.-C. and A.K.; writing—original draft, E.R.; writing—review and editing, A.T.-C. and A.K.; correction of English language, A.T.-C. All authors have read and agreed to the published version of the manuscript.

**Funding:** This research received no external funding.

**Institutional Review Board Statement:** Not applicable.

**Informed Consent Statement:** Not applicable.

**Data Availability Statement:** Data is contained within the article. All data generated or used during the study are available from the corresponding author by request.

**Conflicts of Interest:** The authors declare no conflict of interest.

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
