# Peer review of "The Influence of the Production Process on the Anthocyanin Content and Composition in Dried Potato Cubes, Chips, and French Fries Made from Red-Fleshed Potatoes"

_applsci, doi:10.3390/app11031104_

Round 1

Reviewer 1 Report

This paper has been significantly improved and now warrants publication in Applied Sciences.

Author Response

I am grateful for the review of the article Elżbieta Rytel, Agnieszka Tajner-Czopek, Agnieszka Kita, Agnieszka Tkaczyńska, Alicja Z. Kucharska, Anna Sokół-Łętowska: “The The influence of the production process on anthocyanin content and composition in dried potato cubes, Chips and French fries made from red-fleshed potatoes” (applsci-1086590-1) and valuable suggestions.

General:All corrections in submitted version of the manuscript made by using coloured text (red).According to Reviewer’s suggestions was send to proofread the English language by a native speaker (professionally proofread by Applied Sciences language editing service). I enclose a translation certificate.

Reviewer 2 Report

The revised version of manuscript ID: applsci-1086590 is improved and deserves publication after a minor revision. The study addresses important findings concerning the potatoes of the Herbie 26 variety as matrix for the production of dried potato dice, chips, and French fries due to the optimal chemical composition of tubers, particularly concerning their low reducing sugar content and stability of specific phenolic compounds. The study is an evolution to the existing literature. My specific comments follow the text sequence:

-Line 161. Delete the words’’ The text continues here..’’.

-Line 169. ‘’…in the studied….’’.

-Conclusion

Lines 284-285 must be deleted.

-Graphical abstract

A graphical abstract presenting the experimental procedure and findings would be interesting.

Author Response

              I am grateful for the review of the article Elżbieta Rytel, Agnieszka Tajner-Czopek, Agnieszka Kita, Agnieszka Tkaczyńska, Alicja Z. Kucharska, Anna Sokół-Łętowska: “The The influence of the production process on anthocyanin content and composition in dried potato cubes, Chips and French fries made from red-fleshed potatoes” (applsci-1086590-1) and valuable suggestions.

General:All corrections in submitted version of the manuscript made by using coloured text (red).According to Reviewer’s suggestions was send to proofread the English language by a native speaker (professionally proofread by Applied Sciences language editing service). I enclose a translation certificate. As an answer on the suggestions of 2nd Reviewer:

-Line 161. Delete the words’’ The text continues here..’’- has been deleted

-Line 169. ‘’…in the studied….’’ –  has been corrected

-Conclusion

Lines 284-285 must be deleted –  has been deleted

-Graphical abstract- has been added

This manuscript is a resubmission of an earlier submission. The following is a list of the peer review reports and author responses from that submission.

Round 1

Reviewer 1 Report

The manuscript "The influence of the production process on anthocyanin content and composition in dried potato cubes, Chips and French fries made from red-fleshed potatoes" by Rytel et al. describes how anthocyanin degradation and profiles are influenced in potatoes of the red-fleshed Herbie 26 variety by different methods of processing. The study looks interesting. However, I think the authors should do a comparison study between traditional, light-fleshed varieties and red-fleshed potatoes. As this critical comparison is missing in this manuscript, I suggest that the manuscript can not be accepted in its present form.

Reviewer 2 Report

Introduction: I would be interested in knowing how cubes are consumed. In the USA, I have never heard of dried potato cubes as a consumer product.
71: I believe the authors meant supplied rather than simplified
83: List the manufacturer and location of the slicer
88: List the manufacturer and location of the fryer
95: delete Obtained
110: There is a peculiar un-copyable symbol at the end of this line that should be replace with mm
114: The peculiar un-copyable symbol should be replace with the degree symbol.
115: Include details about the standard curve run on cyanidin 3-O-glucoside and how area counts were converted to concentration.
116: What does “(CygE)/100 g d.m.” mean?
119: What does “marked use” mean? I think the authors mean “analyzed” or “injected”
126: What kind of detector did the UPLC have in addition to the mass spec?
129: Change “nitrogen; dissolution gas” to “dissolution gas, nitrogen”
139: What does “(g/100g f.m.)” mean?
140: The anthocyanin units are mg/100 g, not g/100g
150: Remove “The text continues here”
Figure 2 and 3: What does “Total anthocyanins degrees [%]” mean? The graphs show an increasing loss in anthocyanin content. Reword the legend to reflect this, or change the graphs to show loss of anthoxyanins
263: Remove “This section is not mandatory, but can be added to the manuscript if the discussion is unusually long or complex.”

Reviewer 3 Report

The manuscript investigated the effect of processing on the anthocyanin composition changes of red-colored potatoes. However, the method part lacks of detail and results and discussion part requires in-depth discussion. Additionally, the accurate MS was used; however, the result was presented as unit mass (Table 2). The method requires sample weight used for extraction. Addtinally, vender information is missing. Also there are too many typos throughout the manuscript. Thus, this mansucript cannot be published in current form.